# Seed-Coat Pigmentation Plays a Crucial Role in Partner Selection and N_2_ Fixation in Legume-Root–Microbe Associations in African Soils

**DOI:** 10.3390/plants13111464

**Published:** 2024-05-25

**Authors:** Sanjay K. Jaiswal, Felix D. Dakora

**Affiliations:** Department of Chemistry, Tshwane University of Technology, Arcadia Campus, Pretoria 0183, South Africa

**Keywords:** flavonoids, anthocyanin, pigmentation, soil microbes, disease resistance, nodulation and N_2_ fixation

## Abstract

Legume–rhizobia symbiosis is the most important plant–microbe interaction in sustainable agriculture due to its ability to provide much needed N in cropping systems. This interaction is mediated by the mutual recognition of signaling molecules from the two partners, namely legumes and rhizobia. In legumes, these molecules are in the form of flavonoids and anthocyanins, which are responsible for the pigmentation of plant organs, such as seeds, flowers, fruits, and even leaves. Seed-coat pigmentation in legumes is a dominant factor influencing gene expression relating to N_2_ fixation and may be responsible for the different N_2_-fixing abilities observed among legume genotypes under field conditions in African soils. Common bean, cowpea, Kersting’s groundnut, and Bambara groundnut landraces with black seed-coat color are reported to release higher concentrations of *nod*-gene-inducing flavonoids and anthocyanins compared with the Red and Cream landraces. Black seed-coat pigmentation is considered a biomarker for enhanced nodulation and N_2_ fixation in legumes. Cowpea, Bambara groundnut, and Kersting’s bean with differing seed-coat colors are known to attract different soil rhizobia based on PCR-RFLP analysis of bacterial DNA. Even when seeds of the same legume with diverse seed-coat colors were planted together in one hole, the nodulating bradyrhizobia clustered differently in the PCR-RFLP dendrogram. Kersting’s groundnut, Bambara groundnut, and cowpea with differing seed-coat colors were selectively nodulated by different bradyrhizobial species. The 16S rRNA amplicon sequencing also found significant selective influences of seed-coat pigmentation on microbial community structure in the rhizosphere of five Kersting’s groundnut landraces. Seed-coat color therefore plays a dominant role in the selection of the bacterial partner in the legume–rhizobia symbiosis.

## 1. Introduction

Plants and microorganisms have coexisted for millions of years. At the root–soil interface, plants and microorganisms engage in complex signaling to establish beneficial associations known as symbioses [1]. These interactions are crucial for nutrient acquisition and fitness improvement, and they serve as the foundation for symbiotic relationships, with the most ancient being those formed with nitrogen-fixing bacteria. A key alternative to the use of chemical N fertilizers in cropping systems is biological N_2_ fixation (BNF) [2], a process that has been recognized as the most environmentally friendly and economically sustainable alternative for overcoming soil N infertility [3]. BNF contribution to the N economy of terrestrial ecosystems can be symbiotic, associative, or through free-living microorganisms [4]. However, symbiotic N_2_ fixation in legumes has contributed the most N to global agriculture due to its ability to fix large amounts of biological N in cropping systems for use by the legume and successive crops [5].

Within the broader context of legume–rhizobia symbiosis, seed-coat pigmentation has emerged as a significant factor influencing the establishment of this plant–bacterial association [6]. Though often overlooked, seed-coat pigmentation plays a crucial role in mediating partner selection during the early stages of the legume host–strain symbiosis. Effective nodulation and N_2_ fixation depend on the presence of host plants and compatible nodulating soil bacteria in adequate numbers. Once these conditions are met, legumes can contribute significantly to the N economy of nutrient-poor soils through their ability to establish effective N_2_-fixing symbiosis [7,8]. So far, however, the interplay between seed-coat pigmentation and N_2_ fixation remains relatively unexplored and untapped for sustainable development. This review highlights the role of seed-coat pigmentation in partnership development and N_2_ fixation in legume–rhizobia symbiosis.

## 2. A Historical Perspective on Legume Seed-Coat Pigmentation

From Neolithic agriculture some 12,000 years ago to the present time, grain legumes have continued to play a major role in cropping systems [9]. The evolution of seed-coat pigmentation spans millions of years, reflecting both dynamic environmental pressures and intricate adaptations [10]. Diversity in seed-coat pigmentation is evident across the plant kingdom and reflects the unique ecological niches and adaptive strategies adopted by different plant species. A wide range of seed-coat colors exist within and among legume species [9] (see Figure 1). 

The color of legume seed coats is a trait that is influenced by polyphenolic substances and holds significant evolutionary and agricultural significance. However, crop domestication processes have led to the loss of seed-coat pigmentation, possibly as a trade-off for protection from foragers [11,12]. Wild chickpea (*Cicer arietinum* L.) progenitors, for example, are known to exhibit seed-coat pigmentation that mimics the soil color, thus emphasizing the evolutionary role of seed-coat color [13]. Pigmentation loss has also been linked to the level of bitter-tasting compounds [14].

Legume genotypes have been shown to fix varying amounts of symbiotic N when in association with rhizobacteria. Although the effect of seed-coat color in legume nodulation has been established [6,15,16,17,18], little is known about its influence on the choice of rhizobia by legumes for symbiotic partnership, root nodulation, and N contribution. Darker seed-coat color is reported to contain greater levels of flavonoids, anthocyanins, and anthocyanidins that promote superior symbiotic functioning [17,19]. Hungria and Phillips [17] and dos Santos Sousa [20] have, in fact, shown that common bean (*Phaseolus vulgaris* L.) seedlings generated from a black-seeded genotype formed more root nodules compared with its white isogenic counterpart.

Though known for their esthetics in Africa, legume seed-coat coloration goes beyond mere esthetics as it serves as an adaptive trait with functional significance. The ability of seed coats to exhibit diverse colors is often linked to ecological adaptations such as UV-B protection, signaling mechanisms, and interactions with symbiotic microorganisms. An understanding of the adaptive significance of seed-coat color is likely to unveil its role as a dynamic and strategic trait in the evolutionary arms race between plants and microbes. In the African marketplace, legume accessions are often given local names based on the color of the seed coat [21]. The evolution of the primitive seed-coat color to the sophisticated pigmentations observed in legumes today seems to suggest a historical perspective on the fascinating journey of seed-coat pigmentation.

## 3. Mechanisms of Partner Selection in Symbiotic Interactions

At the molecular level, the pathways linking seed-coat pigmentation to N_2_ fixation remain a focal point of investigation. In the absence of effective indigenous soil bacteria, rhizobial strains must be applied as inoculants for nodulation to occur in legumes. The ability of rhizobia to successfully establish a N_2_-fixing relationship with a legume depends on the symbiotic signals involved in the molecular conversation between the legume and rhizobia during nodule formation (Figure 1) [22]. Usually, the first signals are produced by the plant in the form of flavonoids, which interact with *nod*D proteins of compatible rhizobia. The *nod*D–flavonoid complex then activates the transcription of *nod* boxes that cause the deformation of root hairs, thus allowing entry of the rhizobia into plant roots [22]. 

The flavonoids released by legume seed coats and young roots act as chemo-attractants, cell growth enhancers, and *nod*-gene inducers in symbiotic rhizobia [16,23,24] (Figure 1). Thus, the same flavonoid molecule can therefore elicit different responses in different rhizobia [18]. Unraveling these pathways can provide a molecular framework for understanding the functional role of seed-coat pigmentation in N_2_-fixing symbioses.

## 4. Genetic Determinants of Legume Seed-Coat Pigmentation

The genetic control of seed-coat color involves the influence of specific genes. The stay-green type seed-coat color in chickpea is, for example, caused by loss of function of the gene responsible for chickpea protein “CaStGR1” that affects chlorophyll degradation and retention [25]. In soybean, however, seed pigmentation is tied to the I locus, which affects the expression of chalcone synthase gene and subsequently flavonoid biosynthesis [26]. Similar homologous genes in other legumes, such as the P locus in common bean, Mendel’s A locus in pea, the B locus in chickpea, and the Tan gene in lentil, play major roles in anthocyanin and flavonol biosynthesis [27,28,29,30]. Faba bean presents an intriguing case with two identified homologous genes [31,32].

Recent studies using phenotypic segregation of a ”TU” × ”Musica” of *Phaseolus vulgaris* cross-derived recombinant inbred line population indicated three genes that controlled seed-coat color: one for white (mapped to Pv07) and two for black (mapped to Pv06 and Pv08) [33]. The mapped positions were consistent with classical studies on the V gene and the C locus. Crossing three selected lines validated these regions, confirming that TU alleles resulted in a black phenotype. Furthermore, two genes involved in the anthocyanin biosynthetic pathway were identified as *Phvul.006G018800*, encoding a flavonoid 3′5′hydroxylase, and Phvul.008G038400, encoding the MYB113 transcription factor [33].

Campa et al. [34] also examined 308 common bean (*Phaseolus vulgaris* L.) lines for 10 seed-coat color traits and discovered 31 significant SNP–trait associations (QTNs) that were distributed across 20 chromosome regions. Of these, chromosome Pv08 played a central role, as it influenced phenolic metabolites on chromosomes Pv01, Pv02, Pv04, Pv08, and Pv09 and seed-coat color on chromosomes Pv01, Pv02, Pv06, Pv07, and Pv10. Chromosome Pv08 therefore plays a key role in the phenylpropanoid pathway, with significant effects on common bean seed-coat pigmentation. The genetic basis of the red seed-coat color in cowpea was found to involve two loci, Red-1 (R-1) on chromosome Vu03 and Red-2 (R-2) on chromosome Vu07 [35]. The candidate gene for R-1 was *Vigun03g118700*, encoding dihydroflavonol 4-reductase, which is a cyanidin biosynthesis catalyst, while the candidate gene for R-2 was *Vigun07g118500*, with a nucleolar function and elevated expression in developing seeds. The red color was attributed to the accumulation of cyanidin in the seed coat [35].

## 5. Seed-Coat Pigmentation Is a Major Determinant of Host Plant Choice of Rhizobia for N_2_ Fixation in Legumes

From signaling pathways to nutrient exchange, the genetic determinants of host plant choice of rhizobia can contribute significantly to the success of the symbiotic relationship. Here, seed-coat pigmentation, which is a heritable trait, can contribute to determining the host plant choice of rhizobia in the symbiotic partnership. Common bean, cowpea, Bambara groundnut, and Kersting’s groundnut all exhibit different seed-coat colors that determine the host plant selection of rhizobial partners, a choice that ultimately affects the N_2_-fixing ability of most grain legumes [36,37,38]. Seed-coat color is thus a major determinant of the differences in N_2_ fixation under field conditions, where black-pigmented seeds often produce plants with superior symbiotic abilities. Mohammed et al. [39] recently reported a marked variation in the symbiotic performance and grain yield of cowpea genotypes due to their seed-coat colors, a finding attributed to the ability of the differing seed-coat pigmentations to attract different soil rhizobia. The results of Puozaa et al. [6] showed that despite having similar traits such as growth habits, phenology, flowering and maturity dates, three Bambara groundnut landraces differed markedly in their ability to form nodules and to fix N_2_ due to their seed-coat colors, suggesting their influence on gene expression relating to nodule formation. Further evidence from studies of soybean seed-coat color (e.g., black, brown, and yellow) revealed the role of the isoflavone malonyl-CoA acyltransferase GmMaT2 in the nodulation of this species through modification of the synthesis and secretion of *nod*-gene-inducing isoflavone signals from the seed coat [40].

Mohammed et al. [41] also found that SSR sequence analysis revealed significant genetic diversity among landraces with different seed-coat colors in Kersting’s groundnut (*Macrotyloma geocarpum* Harms). However, due to the unavailability of genetic information specific to Kersting’s groundnut, cowpea SSR markers were used, and the successful cross-genus transferability of SSRs in that study did not only indicate a practical approach for future studies but also suggested an evolutionary closeness between cowpea and Kersting’s groundnut [41]. Furthermore, the nodulation of Kersting’s groundnut by *Bradyrhizobium* strain CB756 [6], which also nodulates cowpea and was initially isolated from *Macrotyloma africanum* Blumenthal and Staples [42], suggested a potential synteny between cowpea and Kersting’s groundnut. A comparison of transcriptomes of contrasting seed-coat colors (black and yellow) and RNA-seq analysis of soybean also discovered the presence of 318 differentially expressed genes involved in ethylene, lipid, brassinosteroid, lignin, and sulfur amino acid biosynthesis in cultivars with black seed-coat pigmentation.

Although the role of seed-coat pigmentation in legume nodulation has been reported before [6,15,17,43,44,45] (see Figure 1), earlier studies of plant nodulation signals failed to recognize the importance of seed-coat color as a determinant of symbiotic success. For example, the very early studies of plant signals involved in nodule formation identified flavonoid molecules such as genistein, genistein-3-O-glucoside, eriodictyol, naringenin, daidzein, and coumestrol in common bean seed and root exudates as *nod*-gene inducers in bean rhizobia [46,47]. Similarly, isoliquiritigenin, genistein, genistein-7-O-glucoside, genistein-7-O-(6-O-malonylglucoside), daidzein, and daidzein-7-O-(6-O-malonylglucoside) were isolated from soybean seed and root exudates as the *nod*-gene inducers for soybean rhizobia [48,49]. The discovered signal molecules created the impression that bacterial *nod*-gene inducers from legumes were mainly flavonoids from the phenylpropanoid pathway. However, a report by Gagnon and Ibrahim [50] showed that aldonic acids (erythronic acid and tectronic acid) from lupin seed and root exudates are the *nod*-gene inducers in lupin rhizobia, including *Rhizobium lupini*, *Mesorhizobium loti*, and *Sinorhizobium meliloti*. This suggested that not all legumes use flavonoids from the phenylpropanoid pathway as *nod*-gene inducers in symbiotic rhizobia, and that there may be undiscovered non-flavonoid molecules that induce *nod*-genes in symbiotic rhizobia.

Furthermore, because these studies were largely conducted using seeds of a single legume variety with a specific seed-coat color, the importance of seed-coat pigmentation in legume–rhizobia symbiosis remained undetected. Except for the study by Hungria et al. [46], which found greater nodulation in seedlings of a black-seeded common bean compared with its cream isogenic counterpart, little is known about the effects of seed-coat color on the nodulation and N_2_ fixation of symbiotic legumes. In that study [46], anthocyanidins were the major *nod*-gene inducers in the seed exudate of the black-seeded common bean, compounds which dominate hugely in the black seed-coat color [51,52,53].

Recent studies have, however, highlighted the importance of seed-coat pigmentation in the nodulation and N_2_ fixation of native African legumes [6,41,54,55]. Black Bambara groundnut landraces were found to show greater nodulation and symbiotic functioning, as evidenced by the lower values of shoot δ^15^N, higher %N from fixation, and greater amounts of fixed N recorded across all study sites [6]. This superior symbiotic performance of the Black Bambara groundnut landrace over its Red counterpart was attributed to its ability to release higher concentrations of *nod*-gene-inducing anthocyanidins as compared to the Red and Cream landraces [51,52,53]. Those results are consistent with the findings of Hungria and Phillips [17], who showed that the black-seeded common bean genotype released higher concentrations of *nod*-gene-inducing flavonoids and thus elicited greater nodulation than its isogenic cream counterpart with lower flavonoid biosynthesis. 

Furthermore, Mbah and Dakora [56] also found that root nodulation, N_2_ fixation, and shoot micronutrient accumulation differed significantly among Bambara groundnut, soybean, and Kersting’s groundnut genotypes with different seed-coat colors. However, applying 5 mM NO_3_^−^ to inoculated seedlings of the three test species reduced plant growth, nodule formation, and nodule dry matter across all seed-coat colors, an indication that nitrate had a negative effect on the signal from the black seed-coat color. It has indeed been previously shown that nitrate can reduce the biosynthesis and release of isoflavone compounds needed for *nod*-gene induction in nodulated soybean [57] and can thus affect nodulation and N_2_ fixation.

Bambara groundnut landraces with diverse seed-coat colors also revealed varied nodulation and plant growth under different field conditions in Cameroon, Burkina Faso, and Nigeria [58,59,60]. Detailed studies are needed to critically evaluate the *nod*-gene-inducing potency of signal molecules released by seeds with differing seed-coat colors, and this should be followed by extensive field evaluation of root nodulation and N_2_ fixation of these test legumes to confirm the link between seed-coat color and symbiotic traits such as *nod*-gene induction, root nodulation, and N_2_ fixation.

## 6. Effect of Seed-Coat Pigmentation on Microbial Colonization

Seed-coat pigmentation is also reported to influence root colonization by both rhizobia and plant-growth-promoting microbes [61,62] (Figure 1). PCR-RFLP analysis of bacterial DNA from root nodules of cowpea and Bambara groundnut landraces (namely, Black, Red, Cream, Red mottled, Black mottled, and Blackeye) grown in Ghanaian and South African soils revealed differences in nodule occupancy based on seed-coat color [54,55]. In essence, seeds differing in seed-coat color attracted different rhizobial strains. Even when cowpea seeds with Black, Red, and Cream pigmentation were planted together in one hole, they attracted diverse bradyrhizobia that clustered differently in the PCR-RFLP dendrogram and occupied distinct positions in the phylogenetic tree [54]. This suggested that the different cowpea and Bambara groundnut seed-coat colors (e.g., Black, Red, Black mottled, and Cream) probably contributed to a soil environment that was conducive to rhizobial richness. In fact, Mohammed et al. [62], Dalamini et al. [63], and Puozaa et al. [6] found that Kersting’s groundnut, Bambara groundnut and cowpea with differing seed-coat colors were selectively nodulated by different rhizobial species. 

This was further confirmed by multilocus sequence analysis performed on rhizobia nodulating Kersting’s groundnut. The results revealed different *Bradyrhizobium* species responsible for nodulating Kersting’s groundnut with differing seed-coat colors [62]. Phylogenetic analysis also placed the test rhizobial isolates in close proximity with different *Bradyrhizobium* species such as *B. vignae* 7-2^T^, *B. subterraneum* 58 2-1^T^, *B. kavangense* 14-3^T^, *B. liaoningense* 2281 (USDA 3622)^T^, *B. yuanmingense* LMG 21827^T^*, B. huanghuaihaiense* CCBAU 23303^T^, *B. pachyrhizi* PAC48^T^, and the type strain of *B. elkanii*, based mostly on seed-coat color [62]. However, in some instances, a significant divergence was found between test isolates and the reference type strains, suggesting that those isolates could be novel *Bradyrhizobium* species [62].

The greater genetic diversity observed among the nodule occupants of the dark-seeded Kersting’s groundnut landraces in this study could be linked directly or indirectly to the higher phenolic compounds produced by the Black landrace compared with the white-seeded Boli landrace [62,64]. Earlier studies have, however, reported higher concentrations of phenolic compounds exuded by seeds with dark seed-coat pigmentation as compared to those with a lighter seed-coat color [51,52], and these molecules are known to play a role in signal exchange during legume–rhizobia symbiosis.

The effect of different seed-coat colors on the microbial community structure in the rhizosphere of five Kersting’s groundnut landraces (namely, Belane Mottled, Boli, Funsi, Puffeun, and Heng Red Mottled) was investigated by Jaiswal et al. [61] using 16S rDNA amplicon sequencing. The results revealed significant selective influences of the landraces on rhizosphere bacteria. The microbial composition and abundance differed significantly among the landraces, with a major landrace effect on some phyla. For example, the rhizosphere of the Belane Mottled landrace was dominated by Proteobacteria, while Bacteroidetes dominated the rhizospheres of the other landraces. Furthermore, except for Puffeun (with a Black seed coat), whose rhizosphere was dominated by *Mesorhizobium*, only *Bradyrhizobium* and *Rhizobium* species of alpha-Proteobacteria were present in the rhizosphere of all the other landraces, although indole-3-acetic–acid-producing Sphingomonas and cellulose-degrading Fibrobacteres were also abundant in the rhizosphere of all landraces [61].

## 7. Evolutionary Dynamics of the Legume–Rhizobia Partnership

Symbiotic relationships between plants and microbes are the outcome of intricate co-evolutionary processes. Grain legumes, including cowpea, soybean, groundnut, and Bambara groundnut, meet a significant proportion of their N requirements from atmospheric N_2_ fixation. Like all grain legumes, seed-coat color variation is strong among Bambara groundnut landraces, and this can range from black to mottled to a lighter cream color. This diversity is primarily attributed to different types of flavonoid compounds and their concentrations present in the seed coat [65]. 

Seed color is often a reflection of the flavonoid compounds present in the seed coat. For example, Bambara groundnut landraces with distinct seed-coat pigmentations were found to contain varying levels of flavonoids and anthocyanins in their seed exudates [6,65]. These phenolic compounds are abundantly present in the seed coat and often act as chemo-attractants for rhizobia, bacterial cell-growth promoters, and inducers of *nod*-genes in symbiotic legumes during nodule formation [16,17,22,66,67].

The presence of a specific legume in the soil can trigger the selection of specific rhizobial groups due to rhizodeposition, which consists mainly of flavonoids released actively or through decomposition by the host plant roots. However, following imbibition during seed germination, legume seeds release water-soluble flavonoid compounds abundantly from the seed coat [22,65,68,69]. Phylogenetic analyses have shown that Kersting’s groundnut is nodulated mainly by *Bradyrhizobium* species [62]. Because of the diverse seed-coat colors of Kersting’s groundnut, this legume tends to attract different microbial profiles in its rhizosphere [61].

## 8. Adapting to Environmental Changes: The Role of Seed-Coat Pigmentation

Environmental changes can pose challenges to symbiotic associations that necessitate adaptive responses. Seed-coat pigmentation plays a key role in the adaptive strategies employed by plants in response to changing environmental conditions. According to Puozaa et al. [55], the cultivation of cowpea with differing seed-coat colors in the low rainfall site of Morwe in South Africa contributed to a higher diversity of cowpea-nodulating rhizobia. Seedling vigor, which is crucial for early crop establishment, may also be influenced by seed-coat color, as evidenced by differences in the seedling vigor among Bambara groundnut landraces with varying seed-coat pigmentation [70]. Despite the superior symbioses induced by dark-seeded grain legumes, site-specific comparisons have shown that the Red Bambara groundnut landrace consistently produced a higher grain yield than the Black and Cream landraces, to the extent that in Cameroon and Tanzania, the Red seed-coat pigmentation is strongly associated with a high grain yield of Bambara groundnut [71,72].

So, while with Bambara groundnut, the Black seed-coat color is linked to higher nodulation and N_2_ fixation [54], the Red seed coat is also strongly associated with greater grain yield [71,72]. More studies are needed to confirm the link between seed-coat pigmentation and legume grain yield. 

Although Bambara groundnut is generally regarded as a drought-tolerant crop, little evidence currently exists to support that claim. However, a recent report by Puozaa et al. [6] seems to suggest a link between black seed-coat pigmentation and plant water-use efficiency in Bambara groundnut. Though they found significant differences in shoot δ^13^C (which is a proxy for water-use efficiency) among the Bambara groundnut landraces tested across different sites, the Black landrace consistently exhibited greater shoot δ^13^C values than the Red and Cream landraces. The higher water-use efficiency of the Black landrace can be attributed to the greater tissue concentration of flavonoids, anthocyanins and anthocyanidins, which are known to not only protect the photosynthetic apparatus under drought conditions but the release could also promote an association with drought-mitigating microbes [73]. Furthermore, a significant correlation was found between shoot δ^13^C and shoot N content, as well as between shoot δ^13^C and soil N uptake [6], suggesting that the enhanced N nutrition from symbiotic fixation by the Black landrace probably contributed to its greater water-use efficiency and drought resistance relative to the Red and Cream landraces. Future studies should focus on finding a link (if any) between seed-coat color and water-use efficiency in grain legumes. Establishing such a linkage would make it easier to breed for water-use efficiency in grain legume foods.

## 9. Grain Legume Seed Coats Are a Natural Source of Nutraceuticals and Anthocyanins

Flavonoids secreted by legume seeds play a major role as signal molecules for attracting compatible rhizobia during symbiotic establishment. As the reproductive unit, seeds carry the genetic material of crop species over time and space, and not only does the seed coat play a vital role in protecting the embryo and/or determining seed dormancy and germination [74,75,76], it also contains a host of novel compounds that are released following the imbibition of water during germination. These molecules include flavonoids, proteins, peptides, amino acids, alkaloids, and terpenoids [65,75]. Flavonoids are phenolic secondary metabolites that confer color on seeds, flowers, and fruits [35]. They play many important roles in plant development, including defense against insect pests and diseases, as well as nodulation in symbiotic legumes (Table 1; Figure 1) [65,77,78,79]. Puozaa et al. [6] also found marked variations in the concentration of phenolic compounds in the seed exudate of Bambara groundnut based on their seed-coat colors. In fact, seeds of the Black landrace released higher levels of flavonoids and anthocyanins, followed by those of the Red landrace, with those of the Cream landrace being the lowest. 

The results of a UPLC-qTOF-MS analysis by Tsamo et al. [51] confirmed these earlier findings of the presence of elevated levels of phenolic compounds in darker-colored Bambara groundnut seeds, the black-seeded adzuki bean, and soybean. Ndakidemi and Dakora [65] also reported higher concentrations of anthocyanins and flavonoids in a black-seeded Bambara groundnut accession as compared to its Cream counterpart. Seed-coat pigmentation therefore plays a crucial role in determining the profile and concentrations of flavonoids, anthocyanins, and anthocyanidins in seeds of cowpea, Bambara groundnut, and Kersting’s groundnut [51,53]. Metabolite profiling by Tsamo et al. [53] also showed that black seeds of legumes are a natural source of nutraceuticals for human nutrition/health, and a reservoir of anthocyanins and anthocyanidins. These flavonoid antioxidants can be exploited for pharmaceutical products, especially since they are implicated in reducing the risk of cancer in humans.

However, consumer preference for cowpea and Bambara groundnut consumption in West Africa, where most of the cowpea in Africa is produced, is influenced by seed-coat pigmentation. Most consumers in West Africa prefer cowpea and Bambara groundnut with a cream or white seed-coat color [80]. This contrasts with Brazil, where “black beans” are highly preferred, possibly due to their higher levels of total phenolics, flavonoids, anthocyanins, anthocyanidins, and antioxidative activity (Table 1) [46,51,52]. Breeding programs for food legumes in West Africa have therefore generally favored varieties/landraces with cream or white seed-coat colors rather than black [81], even though the consumption of these antioxidative compounds has implications for human nutrition and health [82,83].
plants-13-01464-t001_Table 1Table 1Legume seed-coat colors and their impact on plant activities.LegumeSeed-Coat ColorAgronomical EffectReferencesBambara groundnut (*Vigna subterranea* L. Verdc.)BlackEnhanced nodulation and nitrogen fixation[6]Winged bean (*Psophocarpus tetragonolobus*)BrownEnhanced nodulation and nitrogen fixation[84]Common bean (*Phaseolus vulgaris* L.)Light RedDisease resistance and symbiotic nitrogen fixation[85]Soybean (*Glycine max* (L.) Merr.)Black and BrownEnhanced antioxidative activities and anthocyanins[86,87]Soybean (*Glycine max* (L.) Merr.)YellowHigher water absorption[88]Adzuki bean (*Vigna angularis* L.)BlackHigher accumulation of anthocyanins[89,90]Lentil (*Lens culinaris* Medik.)BlackHigher nutraceutical values[19]Peanut (*Arachis hypogaea* L.)Dark redHigher polyphenol content[91]Kersting’s groundnut (*Macrotyloma geocarpum* Harms)BlackHigher nitrogen fixation[41]


## 10. Conclusions

Rhizobia play a pivotal role in agroecosystems, contributing significantly to enhancing overall soil health through their symbiotic relationship with legumes. Numerous reports have indicated that legumes with varied seed-coat colors exhibit diverse responses to N_2_ fixation and sustainable agriculture in African soils. Understanding the molecular and genetic aspects of seed-coat pigmentation is crucial for unraveling symbiotic interactions. Evolutionary dynamics, illustrated by diverse seed-coat colors in legumes like Bambara groundnut and Kersting’s groundnut, showcase the intricate co-evolutionary processes influencing N_2_-fixing partnerships. The connection between seed-coat pigmentation, flavonoid profiles, and rhizobial selection suggests potential applications in optimizing associations with microsymbionts. Further research on neglected legumes, such as Kersting’s groundnut, and the impact of seed-coat colors on soybean seed quality opens up avenues for improved crop productivity and sustainability. 

## Figures and Tables

**Figure 1 plants-13-01464-f001:**
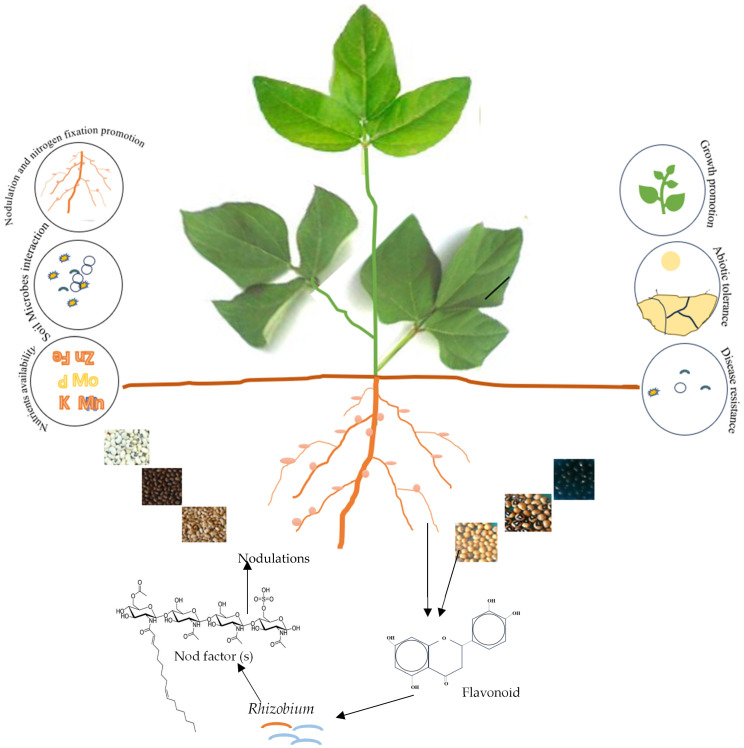
Seed-coat colors and their roles in sustainable agriculture.

## Data Availability

Not applicable.

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
