# Peer review of "Seed-Coat Pigmentation Plays a Crucial Role in Partner Selection and N2 Fixation in Legume-Root–Microbe Associations in African Soils"

_plants, 2024, doi:10.3390/plants13111464_

Round 1

Reviewer 1 Report

Comments and Suggestions for Authors

These authors provide a comprehensive overview of various studies on the topic. It would enhance the manuscript if the authors could pinpoint recent research, highlight current gaps, and outline future prospects, providing readers with additional valuable insights.

The abstract is somewhat lengthy. 

References are needed on line 115. 

In line 392 and throughout the paper, it may be beneficial to discuss the relevance of the mentioned studies to plants and legumes in African soil as has been highlighted in the abstract. 

Author Response

  • These authors provide a comprehensive overview of various studies on the topic. It would enhance the manuscript if the authors could pinpoint recent research, highlight current gaps, and outline future prospects, providing readers with additional valuable insights.

     The abstract is somewhat lengthy. 

    References are needed on line 115. 

    In line 392 and throughout the paper, it may be beneficial to discuss the relevance of the mentioned studies to plants and legumes in African soil as has been highlighted in the abstract. 

    Authors’ Response to Reviewer #1

    • References inserted
    • most of the work on the role of seed coat pigmentation in the legume/rhizobia symbiosis has been done in our Lab. So, there is recent literature to cite.
    • We have highlighted current gaps, and suggested future research (see text highlighted in bold).
    • The length of Abstract has been reduced.

Reviewer 2 Report

Comments and Suggestions for Authors

The authors present an interesting topic - pertinent currently as researchers are deciding how to import N fixing ability  into different microbes 

However  I had trouble reading the paper as it seemed there was continuous repetition of the theme-   it was not apparent how each section varied in the manuscript    I read the same information in several places.

So a major reorganization seems to be required for me.

Also I was after more molecular/biochemical information to provide novel information    not just restatements of older knowledge

The title is generalized  (legume )  versus much of the text that discusses cowpea 

so again there could be better organization 

Many comments are given at the place that is pertinent in the annotated manuscript

Comments on the Quality of English Language

I did not realize that the work was a review  -  was prepared for actual studies on cowpea   rather than many sections  that to me were not  characterized as having different material

Still not sure what the authors are really advocating  for their review  more than just reituration of the role of colors in the seed coats. 

A section on what we do not know yet would be useful to see where this group of facts would be leading.

Author Response

  • The authors present an interesting topic - pertinent currently as researchers are deciding how to import N fixing ability  into different microbes 

    However, I had trouble reading the paper as it seemed there was continuous repetition of the theme-   it was not apparent how each section varied in the manuscript    I read the same information in several places.

    Authors’ Response to Reviewer #2

    • We have deleted repetitions, and merged two sections to avoid repetitions.

    So a major reorganization seems to be required for me.

    Also, I was after more molecular/biochemical information to provide novel information    not just restatements of older knowledge

    The title is generalized  (legume )  versus much of the text that discusses cowpea 

    so again there could be better organization 

    Authors’ Response to Reviewer #2

    • We have deleted repetitions, and merged two sections in the process of reorganization.

    • Most of the work on the role of seed coat pigmentation in the legume/rhizobia symbiosis has been done in our Lab.
    • And we worked on native African legumes (namely, cowpea, Kersting’s groundnut and Bambara groundnut), so it’s not just cowpea.

    Many comments are given at the place that is pertinent in the annotated manuscript

    A section on what we do not know yet would be useful to see where this group of facts would be leading

    Authors’ Response to Reviewer #2

    • Your preference for “successive” in place of “succeeding” crops has been effected.
    • To remove ambiguity elsewhere, we have changed a sentence to read: “Seed color is often a reflection of the flavonoid compound present in the seed coat”.
    • However, because we have re-written sentences in many sections, not all your comments given at the place that is pertinent may have been addressed.
    • We have highlighted current gaps, and suggested future research in different places (see text highlighted in bold).
    • g., the relationship between seed coat color and nodule formation, N2 fixation, water-use efficiency, and even grain yield.
  • Moderate editing of English language required

    Authors’ Response to Reviewer #2

    • The editing and re-organization has improved clarity.

Reviewer 3 Report

Comments and Suggestions for Authors

Dear Authors,

My comments are in pdf-dokument of your manuscript. You will find only minor corrections needed. Nevertheless I would suggest major revision only because I am interested in you response to my comments.

Author Response

My comments are in pdf-dokument of your manuscript. You will find only minor corrections needed. Nevertheless, I would suggest major revision only because I am interested in your response to my comments.

Authors’ Response to Reviewer #3

  • We have re-written some sentences and added new sections, as well as edited out the spaces and empty lines.
  • To avoid any ambiguity, we have changed a sentence to read: “Seed color is often a reflection of the dominant flavonoid compound present in the seed coat”.
  • If the seed coat color is pink, it means the dominant flavonoid in the seed coat has a pink color; and if it’s black, it means the dominant compound in the seed coat is black, as is the case with anthocyanins and anthocyanidins that are found in black-seeded grain legumes.
  • The nodulation gene-inducing flavonoids have different levels of potency. Weak inducers produce fewer nodules with lower nitrogen fixation, while strong inducers produce abundant effective nodules with high levels of nitrogen fixation.

Well if the basis for flavonoid-rhizobia interaction is chemical nature of flavonoids than it is perhaps misleading to use the term pigmentation, which mean different color which in fact suggest electromagnetic nature of interaction. By saying this I am interested is there any evidence that initial rhizobia-seed interaction is governed by electromagnetic interaction instead of chemical? For instance is there an experiment which has tested interaction of the rhizobia with painted seeds.

I am asking this because bacteria can percept electromagnetic radiation and also I recently read a paper which show that cyanobacteria growth was influenced by the color of the solid substrate at which they were growing. Maybe rhizobia function in similar way.

My point is that you should refer to the flavonoid which is dominant in some seed or to the dominant group of compounds and not to the color itself (if color does not have anything to do with microbe-seed interaction). I mean the same seed color could be achieved by different compounds.

Authors’ Response to Reviewer #3

  • The pigmentation is referring to the color of the dominant compound in the seed coat, which may be potent thus producing many nodules and fixing high levels of nitrogen; or less potent with lower nodulation and less nitrogen fixation.
  • Basically, the relationship is chemical in nature, based on perception and recognition of molecules produced by each partner.
  • It is the color that identifies the molecule.
  • No, the same seed color cannot be achieved by different compounds.

Round 2

Reviewer 2 Report

Comments and Suggestions for Authors

The  review features on a significant feature of the functions of seed coats

the work reads well  -   you could read through though and add  words to indicate what legume you are talking about  for all the examples you go through 

there are some sticky notes that indicating  potential modifications  

Author Response

Authors' response: We have reviewed all the sticky notes and made the suggested minor modifications in the revised manuscript.

Reviewer 3 Report

Comments and Suggestions for Authors

-

Author Response

Thank you so much